# Chlorantraniliprole Residual Control and Concentration Determination in Cotton

**DOI:** 10.3390/insects14020176

**Published:** 2023-02-10

**Authors:** Jacob Smith, Whitney D. Crow, Angus L. Catchot, Donald R. Cook, Jeffrey Gore

**Affiliations:** 1Delta Research and Extension Center, Mississippi State University, Stoneville, MS 38776, USA; 2Department of Biochemistry, Molecular Biology, Entomology and Plant Pathology, Mississippi State University, Mississippi State, MS 39762, USA

**Keywords:** bollworm, diamides, insecticide, pest management

## Abstract

**Simple Summary:**

Data collected from studies in 2020 and 2021 determined that concentrations of chlorantriliprole were detected in the leaves up to 28 days after treatment, were detected in petals up to 14 days after treatment, and were not detected in the anthers. Bollworm mortality was determined on leaves, petals, and anthers, with feeding in the field tending to be concentrated on fruiting structures. While bioassay results varied, field results could also be variable in terms of feeding location.

**Abstract:**

Studies were conducted in 2020 and 2021 at the Delta Research and Extension Center in Stoneville, MS, to determine the residual concentrations of chlorantraniliprole in cotton (*Gossypium hirsutum*, L.) leaves, as well as the concentrations in petals and anthers that developed after the time of application. Foliar applications of chlorantraniliprole were applied at four rates for leaves and two rates for petals and anthers at the second week of bloom. Additional bioassays were conducted to determine mortality of corn earworm (*Helicoverpa zea*, Boddie) in anthers. For the leaf study, plants were partitioned into three zones consisting of top, middle, and bottom zones. Leaf samples from each zone were analyzed for chemical concentrations at 1, 7, 14, 21, and 28 days after treatment (DAT). Residual concentrations, although variable, persisted through all sampling dates, rates, and zones tested. In this study, chlorantraniliprole remained detectable up to 28 DAT. Results from the cotton flower petal and anther studies detected concentrations of chlorantraniliprole in petals at 4, 7, 10, and 14 DAT, but no concentrations were detected in anthers. Therefore, no mortality of corn earworm was recorded in the anther bioassays. A series of diet-incorporated bioassays were conducted using concentrations previously found in the petal study to determine baseline susceptibilities of corn earworms and predicted mortality. Results from the diet-incorporated bioassays showed similar susceptibility in field and lab colony corn earworms. Concentrations of chlorantraniliprole could provide up to 64% control of corn earworm when feeding occurs on the petals.

## 1. Introduction

Corn earworm, *Helicoverpa zea*, (Boddie), is an economically important pest of both soybean and cotton in the Mid-South. If left uncontrolled, corn earworm can be detrimental to cotton production systems. Behind the tarnished plant bug (*Lygus lineolaris* (Palisot de Beauvois)), corn earworm was the second most economically damaging pest of Mississippi grown cotton in 2020 [1]. Small corn earworm larvae feed on young fruiting structures near the oviposition site and move to maturing fruiting structures as the insect molts to later instars [2]. Preferred feeding structures include flowers and wilting flower corollas (bloom tags), which can lead to abscission of fruiting and/or a reduction in yield [3,4,5]. Planting of transgenic Bt technology is the primary management practice used in cotton to control corn earworm infestations, but due to Bt resistance, supplemental foliar insecticides may be needed [6,7,8,9,10].

The use of insecticides targeting corn earworm has become even more prominent in recent years due to documented failures of two-gene Bt cotton and insecticide resistance [6,7,8,9,10]. Chlorantraniliprole (Prevathon^®^ FMC Corporation, Philadelphia, PA, USA), a diamide insecticide, was released in 2008. This insecticide has proven to be very effective at controlling lepidopteran pests such as corn earworm, fall armyworm, and soybean looper [11,12,13]. Research has shown systemic and translocative movement of chlorantraniliprole along with a long residual [13,14,15,16]. When applied to the soil, the insecticide was taken up by the root systems of cabbage or other plants and moves upwards in the plant’s xylem [15]. Research by Adams et al. [13] found that when chlorantraniliprole was foliar applied, it moved systemically to vegetation in soybean, but no concentrations were detected in reproductive structures. Adams et al. [13] also conducted bioassays targeting corn earworm in soybean and discovered when soybeans are infested at R1-R3, the systemic efficacy of chlorantraniliprole can provide some control in the foliage. According to field bioassays conducted by Babu [16], chlorantraniliprole had lethal and sublethal effects on corn earworm feeding on fresh cotton leaf tissue for up to 22 days. Limited research has been conducted on the systemic activity of chlorantraniliprole in fresh flowering structures of cotton or soybean [16]. The objective of this study was to determine the residual concentrations of chlorantraniliprole in cotton leaves, as well as petals and anthers that developed after the time of application.

## 2. Materials and Methods

### 2.1. Field Experiment Details

Three field studies were conducted in 2020 and 2021 at the Delta Research and Extension Center in Stoneville, MS. All field trials were conducted with the same plot layout and experimental design. Non-Bt cotton (Deltapine 1822XF, Bayer CropScience, St. Louis, MO, USA) was planted on raised beds at the Delta Research and Extension Center in Stoneville, MS, between 8 and 20 May in 2020 and 2021. Plots were 6 rows wide with 1.02 m row spacing and were 12.2 m long. Plots were separated by fallow alleys that were 3.04 m long. All standard production practices were implemented according to Mississippi State University Extension Service recommendations. Insect pests were managed on the basis of economic thresholds [10] using insecticides without lepidopteran activity. Field studies were arranged as a randomized complete block design with four replications. In order to minimize the risk of drift between plots, only the center four rows were sprayed. Canopeo (Mathworks, Inc., Natick, MA, USA), a mobile device application developed to measure green canopy coverage, was used to determine >90% canopy closure in all plots one day prior to application. When handling treated material for all field tests, gloves were worn and changed between plots to reduce the risk of insecticide contamination. Samples were kept frozen at −18 °C until samples could be transported to the Chemical Analysis Laboratory at Mississippi State University.

### 2.2. Cotton Leaf Application

This study was conducted in 2020 to determine residual concentrations of foliar applied chlorantraniliprole in cotton leaves throughout the plant canopy. Prior to insecticide application, 10 plants per plot were sampled to calculate the average number of nodes. The data were used to determine the partitioning of zones. These zones consisted of a top (16th node), middle (13th node), and bottom (8th node) zone. Plants were partitioned accordingly to determine the distribution of insecticide throughout the canopy to better estimate potential insect mortality. Four rates of chlorantraniliprole (0.028, 0.053, 0.078, 0.103 kg ai ha^−1^) and an untreated control were applied to plots at the second week of bloom with a John Deere 6000 Hi clearance sprayer (John Deere, Moline, IL, USA) calibrated to deliver 93.5 L ha^−1^ at 350 kPa through TX-6 ConeJet^®^ VisiFlo^®^ Hollow Cone Spray Tip nozzles (2 nozzles per row) (TeeJet^®^ Technologies, Glendale Heights, IL, USA).

At 1, 7, 14, 21, and 28 days after treatment (DAT), 15 cotton plants from the center two rows of each plot were clipped at the base of the soil line and bundled together with cable ties. Each bundle was placed in 113 L trash bags (Hefty, Reynolds Consumer Products LLC, Lake Forest, IL, USA) and transported to the laboratory. Within each zone, 15 leaves were removed and placed in 946 mL self-sealed plastic bags (Ziploc, S. C. Johnson and Son, Inc., Racine, WI, USA) and placed in the freezer. Leaf bioassays were conducted concurrently with the previous test. In the laboratory, 10 and 15 mm cotton leaf disks were placed into 59.2 mL cups from each plot (Solo^®^, Dart Container Corp., Mason, MI, USA). Lids were placed upside down on the table and filled with 1% water agar (Product No. 7060, Frontier Agricultural Sciences. Newark, DE, USA) solution to prevent desiccation. One second instar lab colony corn earworm larva was placed in each cup, and lids with agar were attached. Mortality was recorded 4 days after infestation (DAI), and larvae were recorded dead if they could not right themselves when rolled on to their dorsal surface.

### 2.3. Cotton Flower and Anther Application

In 2020 and 2021, studies were conducted to determine chlorantraniliprole concentrations and mortality of corn earworm in cotton petals and anthers that were undeveloped prior to the application. Treatments consisted of two rates of chlorantraniliprole (0.053 and 0.078 kg ai ha^−1^) and an untreated control. Applications were made at the second week of bloom.

At 4, 7, 10, and 14 DAT, forty-five cotton flowers per plot were removed from the upper one-third of plants in the center two rows. Samples were placed in 946 mL self-sealed plastic and transported back to the laboratory. Bracts were removed from each flower, and fifteen flowers with anthers were placed in clean bags for chemical analysis. Anthers were removed from the remaining flowers. Twenty anthers were placed in separate clean bags for the additional chemical analysis. The other ten anthers were used for bioassays with corn earworm larvae.

### 2.4. Insect Rearing

Laboratory-reared (lab colony) corn earworms, originating from larvae collected from non-Bt corn in 2006, were maintained at the Mississippi State University insect rearing facility. Wild individuals were added to the colony on a biannual basis to maintain genetic diversity within the colony. This colony was reared under recommended conditions of 25 °C, 80% relative humidity, and 16:8 (L:D) photoperiod. Wild (field colony) corn earworms were collected 15 May 2021 from crimson clover, *Trifolium incarnatum* L., near Vicksburg, Mississippi, and placed into 59.2 mL cups (Solo^®^, Dart Container Corp., Mason, MI, USA) containing Stonefly *Heliothis* diet (Product No. 38-0600, Ward’s Science, Rochester, NY, USA) with matching lids. Larvae from the field were maintained in a climate-controlled room set to 26.7 °C, 80% humidity, and (L:D) photoperiod of 16:8 h at the Mississippi State University Delta Research and Extension Center insect rearing facility in Stoneville, MS. At pupation, approximately 40 pupae from each colony were placed in 3.79 L cardboard buckets and covered with cheesecloth, which acted as a detachable oviposition location for moths. Cheesecloth with eggs were placed into 3.79 L self-sealing bags. Once neonates appeared, approximately 100 larvae were transferred into 473 mL plastic deli containers (Fabr-Kai Corp, Kalamazoo, MI, USA) filled with a thin layer of Stonefly *Heliothis* diet (Product No. 38-0600, Ward’s Science, Rochester, NY, USA) and covered. Transferred larvae were placed back into the climate-controlled rearing room until larvae reached second instar when they were used for bioassays.

### 2.5. Cotton Anther Bioassay

Bioassays were conducted in 2021 at the Mississippi State University Delta Research and Extension Center in Stoneville, Mississippi, to determine the efficacy of chlorantraniliprole in cotton anthers. Ten anthers from each of the four reps for a total of forty per treatment were placed individually into 59.2 mL plastic cups (Solo^®^, Dart Container Corp., Mason, MI, USA). One second instar lab colony corn earworm larva was placed in each cup, and matching lids were attached. Mortality was evaluated 2 days after infestation, and larvae were recorded dead if they could not right themselves when rolled on their dorsal surface.

### 2.6. Diet-Incorporated Bioassays

Diet-incorporated concentration-mortality bioassays were conducted in 2021 at the Mississippi State University Delta Research and Extension Center in Stoneville, Mississippi, to compare the susceptibility of lab and field colony corn earworms to concentrations of chlorantraniliprole found in the cotton flower chemical analysis study. The methods for preparation of the diet-incorporated bioassays were similar to Temple et al. (2009). A commercial formulation of chlorantraniliprole (Prevathon^®^; 5 SC; 41.5 g ai/L, FMC Corporation, Philadelphia, PA, USA) was used for this study. Dilutions were prepared by adding 0.1 g of formulated chlorantraniliprole to 1000 mL of distilled water. Serial dilutions of the desired concentrations of chlorantraniliprole were diluted in 200 mL of distilled water. Five concentrations (5, 25, 75, 100, 125 PPB) and an untreated check were combined with Stonefly *Heliothis* Diet (Product No. 38-0600, Ward’s Natural Science, Rochester, NY, USA) to yield 200 mL of insecticide-treated diet for each concentration. To distribute insecticide evenly, the treated diet was hand-agitated for 60 s, and gloves were worn and changed between concentrations to reduce the risk of cross contamination. A total of 28.35 g of treated diet was placed into 59.2 mL cups (Solo^®^, Dart Container Corp., Mason, MI, USA) to yield 20 cups per treatment. One second instar corn earworm larvae was infested per cup. Cups were covered with matching lids and placed into a rearing chamber maintained at 26.7 °C, 80% humidity, and a light/dark cycle of 16:8 h. Insects were evaluated 5 days after infestation (DAI) for mortality. Intoxicated larvae were recorded dead if they could not right themselves when rolled to their dorsal surface. Lab colony corn earworms were replicated eight times (160 larvae per concentration), while field colony corn earworms were replicated three times (60 larvae per concentration).

### 2.7. Chemical Analysis

Cotton leaf, flower, and anther samples were analyzed using a modified QuEChERS by LC–MS/MS and GC–MS/MS procedure developed by Anastassiades and Lehotay [17], and results were displayed in parts per billion (PPB) of active ingredient. Leaf, flower, or anther samples were ground into a powder, and 5 g of the sample was deposited into a 50 mL polypropylene tube. A total of 5 g of clean, lab-grown samples were placed into two 50 mL polypropylene tubes for “blank” and “spike” samples. Spike samples were given adequate concentrations of insecticides to be tested to ensure concise readings and the blank sample was left clean. Ceramic beads were placed in each tube for homogenizing the samples when centrifuging. Additionally, 10 mL of high-performance liquid chromatography water was deposited in the tubes. A GenoGrind (SPEX Sample Prep, Metuchen, NJ, USA) plant tissue homogenizer was used to centrifuge all samples at 1000 RPM for five minutes. Following the first round of centrifuging, each sample received 10 mL of acetonitrile (ACN), which allows for the extraction of the active ingredient, and were centrifuged again for five minutes. MgSO_4_ (anhydrous magnesium sulfate) was then added to samples to separate the active ingredient from plant material. An additional five minutes of centrifuging was needed to separated water and ACN. Samples were placed back into the GenoGrind, and centrifuging time and RPM were increased to ten minutes and 4000, respectively. Following this final round of centrifuging, complete separation of the mixture was achieved with the top layer of liquid containing the residual active ingredient. A total of 1 mL of the extracted liquid was placed into 15 mL polypropylene tubes. Tubes containing the extracted liquid were placed into an auto sampler vial with a PTFE/PVDF filter and analyzed using a LC–MS/MS or GC–MS/MS for GC-amenable pesticides. Recovery of residual insecticide ranged between 85 and 101% (mostly >95%) (Anastassiades and Lehotay 2003) [17].

### 2.8. Data Analysis

Trace amounts of drift in the untreated controls were observed in the cotton leaf chemical analysis study and were omitted from statistical procedures. Chemical analysis data were transformed using log transformation prior to statistical analysis, and non-transformed means and standard errors were reported. The analysis for the cotton leaf study included zone, rate, DAT, and their interactions were considered fixed effects in the model. For the cotton flower chemical analysis study, rate, DAT, and their interaction were considered fixed effects in the mode. Replication was established as the random effect for all studies. Chemical analyses for cotton leaves, petals, and anthers were analyzed with a mixed model analysis of variance (PROC GLIMMIX, SAS 9.4, SAS Institute Inc. Cary, NC, USA). The Kenward–Roger method was used to calculate degrees of freedom. Means and standard errors were calculated using the PROC MEANS statement. LS means were separated using Fisher’s Protected LSD α = 0.05. Results of concentrations found in cotton flowers were used for diet incorporated bioassays.

Data from diet-incorporated bioassays were analyzed using probit analysis in SAS 9.4 (PROC PROBIT, SAS Institute Inc., Cary, NC, USA) to calculate LC_50_ values (PPB). LC_50_ values were considered different when 95 percent confidence intervals did not overlap. The regression equation was determined from the cotton petal chlorantraniliprole concentrations, which is a function of application rate and sample timing (DAT). For the LC_50_ regression equation, concentration and mortality were used to determine the percent predicted mortality at the various sample dates.

## 3. Results

### 3.1. Diet-Incorporated Bioassays

Similar LC_50_ values were observed between field colony (LC_50_ = 30.1 PPB) and lab colony (LC_50_ = 30.0 PPB) corn earworms in the diet-incorporated bioassays. Confidence intervals overlapped among the two populations and were considered not significantly different (field colony (8.9–56.03) and lab colony (10.8–56.80)). Since no differences in responses between colonies were observed, lab colony corn earworms were used for the remainder of bioassays.

### 3.2. Chemical Analysis of Cotton Leaves

There was not a significant interaction between rate and DAT for the top (F = 0.23; df = 12, 37; *p* = 0.1), middle (F = 1.1; df = 12, 38; *p* < 0.01), and bottom (F = 0.59; df = 12, 37; *p* < 0.01) zones of the plant for chlorantraniliprole concentrations. However, the main effects of rate were observed for concentrations in the top, middle, and bottom zone of the plant (Table 1).

In the top zone, concentrations detected with an 0.078 and 0.103 kg ai ha^−1^ rate were similar but significantly higher than what was found with the 0.028 and 0.053 kg ai ha^−1^ rate (F = 8.3; df = 3, 37; *p* < 0.01). Additionally, 0.028 and 0.053 kg ai ha^−1^ rates recorded similar chlorantraniliprole concentrations for leaves in the top zone. For leaves in the middle zone, concentrations declined as rates were reduced (F = 33.7; df = 3, 38; *p* < 0.01). Mean concentrations for the lowest use rate (0.028 kg ai ha^−1^) were significantly lower compared to all other rates in the middle of the plant. Differences in concentrations for rates in the bottom zone were observed (F = 3.4; df = 4, 37; *p* = 0.03). Concentrations associated with the 0.028 and 0.053 kg ai ha^−1^ rates were not significantly different but were lower than the higher use rates (0.103 and 0.078 kg ai ha^−1^) in the bottom of the plants.

Additionally, the main effects of DAT were significant for concentrations in the top of the plant (F = 12.9; df = 4, 37; *p* < 0.01) (Table 1). Significantly higher concentrations were recorded at 1 DAT compared to all other sampling dates in the top zone. At 28 DAT, concentrations were reduced by 98% compared to concentrations at 1 DAT. No differences in concentrations were recorded between 21 and 28 DAT in the top zone. Concentrations in leaves in the middle zone were significantly different from each other and decreased over time (F = 91.1; df = 4, 38; *p* < 0.01). Although chlorantraniliprole was present throughout all sampling dates, concentrations decreased 96% from 1 to 28 DAT in the middle of the plant. At 1 DAT, concentrations in the bottom zone were higher compared to all other sampling dates (F = 10.8; df = 4, 37; *p* < 0.01). Reduced concentrations were recorded at 21 and 28 DAT compared to 7 DAT.

### 3.3. Chemical Analysis of Cotton Petals and Anthers

The interaction between rate and DAT (F = 2.1; df = 3, 14; *p* < 0.15) was not significant for chlorantraniliprole concentrations in cotton petals. However, the main effects of rate (F = 12.4; df = 1, 14; *p* < 0.01) and DAT (F = 92.4; df = 3, 14; *p* < 0.01) were observed. For rate, concentrations were lower when applied at a rate of 0.053 kg ai ha^−1^ compared to a rate of 0.078 kg ai ha^−1^ (Table 2).

Additionally, concentrations detected at 4, 7, 10, and 14 DAT were significantly different from each other and decreased over time. However, no chlorantraniliprole concentrations were detected when anthers were tested alone.

## 4. Discussion and Conclusions

Corn earworm control is largely dependent on Bt cotton varieties, but due to resistance issues, foliar applications of insecticides are often needed [6,7,8,9,10]. The use of insecticides in today’s world emphasizes effectiveness against target pests, effects on beneficial insects, mammalian toxicity, and long residuals to combat resistance and limit pesticide applications. Diamides, including chlorantraniliprole, showed low activity on non-target beneficial insects and reduced mammalian toxicity [18,19]. Additionally, chlorantraniliprole is active against lepidopteran pests and provides effective residual control of caterpillar species in crops such as cabbage, soybean, and cotton [13,15,16]. On the basis of the cotton leaf chemical analysis study, concentrations of foliar applied chlorantraniliprole remained in or on leaves for up to 28 DAT. Generally, as sampling date increased and rate decreased, concentrations decreased. Concentrations at 1 DAT were relatively high in leaves in the top of the plant compared to other sampling dates, more than likely due to good spray coverage in the uppermost zone in the canopy. In a study in snap beans, residual activity decreased over time as the compound degraded and diluted through the plant [20]. In this study, in cotton, environmental factors such as rainfall events and degradation possibly contributed to concentrations in the top of the plant decreasing rapidly over time. Leaves from the bottom of the canopy had lower concentrations of chlorantraniliprole compared to leaves in the top and middle zone, possibly a result of poor spray coverage deep in the canopy (>90% canopy closure). Residual concentrations of chlorantraniliprole could be dependent on plant size and canopy closure at the time of applications. Nonetheless, chlorantraniliprole concentrations, although variable, persisted through all sampling dates, rates, and zones tested. Data from cotton leaf bioassays conducted concurrently with the chemical analysis of leaves were unusable due to high untreated control mortality. However, nearly all concentrations of chlorantraniliprole detected in the cotton leaf chemical analysis were in greater concentrations than what was used in the diet incorporated assays where mortality occurred.

Typically, corn earworm feeding is of most concern to marketable structures of cotton such as squares and bolls. In a study by Braswell et al. [21], leaves of cotton plants were the most attractive for corn earworm oviposition, and many eggs were found on leaves deep in the canopy. Additionally, upon hatching on leaves, first instar corn earworm larvae were observed feeding on fruiting structures near the oviposition site [21]. With the known preferred oviposition and feeding site documented and the lasting residual concentrations of chlorantraniliprole detected throughout the canopy in this study, this compound shows potential to cause residual mortality up to 28 DAT of corn earworms migrating from leaves to fruiting structures. On the basis of this study, chlorantraniliprole should continue to be used in lepidopteran insect pest management in cotton, and long residual effects could be expected.

Chlorantraniliprole appears to be absorbed through the stem in soybean and transported through the xylem [13,14]. Since chlorantraniliprole was applied to entire plants with immature flower buds present, we cannot confidently assume concentrations in cotton flower petals were systemic. On the basis of research by Adams et al. [13], chlorantraniliprole moved systemically to vegetation in soybean and provided some control of corn earworm, but no mortality was recorded in reproductive structures. Results from the cotton anther study indicated no concentrations of chlorantraniliprole in anthers, and no mortality of corn earworm was recorded. Overall, concentrations of chlorantraniliprole were detected in cotton flower petals but not in anthers. In this study, in cotton, concentrations of chlorantraniliprole ranged from 0 to 50.9 parts per billion (PPB) in flower petals out to 14 DAT. However, on the basis of field observations, corn earworm feeding is almost always exclusive to the cotton anthers and not the petals themselves because survival rates are generally greater on anthers than petals [5]. Additional studies were conducted to determine mortality of corn earworm using the diet-incorporated bioassay method. Similar LC_50_ values for both lab and field colony corn earworms suggest susceptibility is comparable in the two tested populations. Since no statistical differences in LC_50_ values were observed, lab colony corn earworms were used for the remaining assays. Some corn earworm control due to chlorantraniliprole might be excepted up to 14 DAT in flower petals. On the basis of this study, bollworm mortality up to 47% might be expected in cotton flowers undeveloped at the time of a chlorantraniliprole application, assuming feeding was occurring on the petals and not anthers (Figure 1).

Insecticides, primarily diamides, are a primary management option for corn earworm l in cotton but now serve as a supplemental control in combination with Bt cotton varieties [9,10]. However, flowering structures are most susceptible to corn earworm feeding in Bt cotton [22]. The use of Bt cotton varieties supplemented by diamide insecticide applications should continue, and increased control might be expected in flowering structures due to the possible systemic nature of chlorantraniliprole.

## Figures and Tables

**Figure 1 insects-14-00176-f001:**
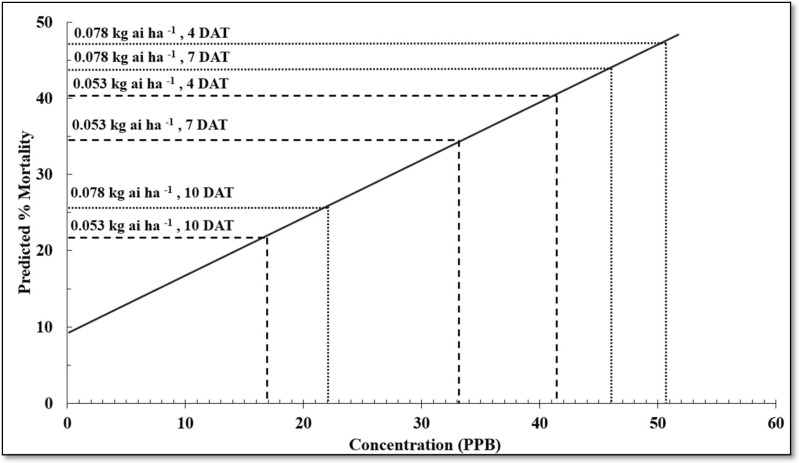
Relationship between chlorantraniliprole concentration and corn earworm mortality in diet-incorporated bioassays and predicted mortality based on concentrations in cotton flower petals.

**Table 1 insects-14-00176-t001:** Mean (SEM) of chlorantraniliprole concentrations (PPB) by zone in cotton leaves in the top, middle, and bottom of the plant to determine the residual activity throughout the canopy in the 2020 study.

Rate ^1^	Top ^2^	Middle	Bottom
	Mean (±S.E.)		
0.028	1067 b (452)	817 d (281)	396 b (147)
0.053	1227 b (625)	1473 c (446)	594 b (203)
0.078	2661 a (938)	1787 b (491)	1038 a (247)
0.103	3879 a (1305)	2874 a (772)	1480 a (500)
F	8.3	33.7	3.4
d.f.	3, 37	3, 38	4, 37
*p* > F	<0.01	<0.01	0.03
**DAT ^3^**			
1	7471 a (1406)	4815 a (759)	2303 a (519)
7	2253 b (404)	1882 b (352)	986 b (275)
14	1001 bc (251)	1345 c (267)	663 bc (242)
21	302 cd (82)	450 d (99)	318 cd (78)
28	102 d (34)	196 e (41)	148 d (37)
F	12.9	91.1	10.8
d.f.	4, 37	4, 38	4, 37
*p* > F	<0.01	<0.01	<0.01

Letters were assigned on the basis of log-transformed statistics. Means within a column followed by the same letter were not significantly different according to Fisher’s protected LSD (α = 0.05). Means and standard errors are expressed as concentrations (PPB) of chlorantraniliprole. Trace amounts of drift were detected in untreated plots. ^1^ Rate of chlorantraniliprole expressed in kg ai ha^−1^. ^2^ Plants were partitioned into three “zones” consisting of leaves from a top (16th node), middle (13th node), and bottom (8th node) zone. ^3^ DAT = days after treatment.

**Table 2 insects-14-00176-t002:** Mean (SEM) of chlorantraniliprole concentrations (PPB) detected in cotton flower petals.

Rate ^1^	Mean (±S.E.)
0.078	25.5 b (4.0)
0.103	31.8 a (5.9)
F	12.4
d.f.	1, 14
*p* > F	<0.01
**DAT ^2^**	
4	46.4 a (2.9)
7	39.3 b (3.1)
10	20.2 c (1.8)
14	8.7 d (1.1)
F	93.4
d.f.	3, 14
*p* > F	<0.01

Letters were assigned on the basis of log-transformed statistics. Means within a column followed by the same letter were not significantly different according to Fisher’s protected LSD (α = 0.05). Means and standard errors are expressed as concentrations (PPB) of chlorantraniliprole. ^1^ Rate of chlorantraniliprole expressed in kg ai ha^−1^. ^2^ DAT = days after treatment.

## Data Availability

The data presented in this study are available in the article.

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
