# Peer review of "Chlorantraniliprole Residual Control and Concentration Determination in Cotton"

_insects, 2023, doi:10.3390/insects14020176_

Round 1

Reviewer 1 Report

I have carried out review of the article “Chlorantraniliprole residual control and concentration determination in cotton.” by Smith et al.

I attest the study was well conducted, and quite interesting. In addition, the manuscript was well written and the organization is of good standard. The experimental procedures were as well clearly highlighted. I do not have any reservation against the acceptance of the manuscript for publication in the insect journal.

However, I have a few suggestions

1. The authors might consider revising the keywords, as some of the keywords: “cotton; chlorantraniliprole”, are already contained in the paper title.

2.      Amend minor errors in the body of the text as follows;

L94-95 – “freezer (Ziploc, S. C. Johnson & Son, Inc., Racine, WI).” is wrong. Should be manufacturer’s details for the refrigerator instead. Please, delete “(Ziploc, S. C. Johnson & Son, Inc., Racine, WI).” from here and move” to after “946mL self-sealed plastic bags”.

In addition, I assume, there is no need to repeat the manufacturer details at every mention of self-sealed plastic bag all through the manuscript, see L111, L130, one time mention is okay, maybe.

L287 – Do you mean “up to 28 DAT”???

L313 – “and long residual effects could be expected”???

L339-340 – Delete "control" after earworms. Revise as “are a primary management option for corn earworm in cotton”. 

3.      I am a little bit confused why the authors decided to include Fig. 1 under discussion (L335), instead of under result section.

Author Response

Keywords changed 

L94-95 – “freezer (Ziploc, S. C. Johnson & Son, Inc., Racine, WI).” is wrong. Should be manufacturer’s details for the refrigerator instead. Please, delete “(Ziploc, S. C. Johnson & Son, Inc., Racine, WI).” from here and move” to after “946mL self-sealed plastic bags”. Corrected

In addition, I assume, there is no need to repeat the manufacturer details at every mention of self-sealed plastic bag all through the manuscript, see L111, L130, one time mention is okay, maybe. Corrected

L287 – Do you mean “up to 28 DAT”??? Corrected

L313 – “and long residual effects could be expected”??? Corrected

L339-340 – Delete "control" after earworms. Revise as “are a primary management option for corn earworm in cotton”.  Corrected 

Reviewer 2 Report

Systemic and translocative movement of Chlorantriniliprole and residual analysis in cotton done in this study. A very well conducted study enclosing multiple aspects of cotton and corn earworm over two cropping season. The study is the need of the time where pests developing high resistance in multi-cropping area and pattern (cotton, corn, soya, wheat) where multiple hosts are available for pest to continue life cycle to overcome pesticide hurdles. Few very minor things need to be added before final publishing. 

Line 66: please mention plant to plant distance.

Line 99: the sentence gives an impression that only one replicate was conducted. If not please mention. (I could not see the information in the paragraph lines from 90-100.)

Line 159: How much diet (weight) was added per cup?

General comment; In such studies, a correlation is maintained/discussed between residues and MRL values (maximum residual limits) set by EPA/industry. If you think it's appropriate to discuss it (not going off focus the study) then just mention the MRL values of Chlorantriniliprole in discussion to shed a clear understanding, although you have mentioned at line 282. FMC will have those values or PHI  (pre-harvest interval) value can serve the purpose as well.

References: # 9, line 376, please abbreviate the name of journal as 'Environ. Entomol."

Ref. # 16: Line 395. please add full stop after Manag. Do the same for ref. 19 line 404.

Ref. #20, line 406, please abbreviate protection as 'Protect.

Author Response

Line 66: please mention plant to plant distance. Row spacing was included. Plant spacing within the row? A standard seed treatment rate was used. 

Line 99: the sentence gives an impression that only one replicate was conducted. If not please mention. (I could not see the information in the paragraph lines from 90-100.) Updated to be more clear. 

Line 159: How much diet (weight) was added per cup? Added

References: # 9, line 376, please abbreviate the name of journal as 'Environ. Entomol." Corrected

Ref. # 16: Line 395. please add full stop after Manag. Do the same for ref. 19 line 404. Corrected

Ref. #20, line 406, please abbreviate protection as 'Protect. Corrected